# ONE-SHOT MULTI-LABEL CAUSAL DISCOVERY IN HIGH-DIMENSIONAL EVENT SEQUENCES

## ABSTRACT

Understanding causality in event sequences where outcome labels such as diseases or system failures arise from preceding events like symptoms or error codes is critical in domains such as healthcare, cybersecurity, and vehicle diagnostics. Yet, existing causal discovery methods struggle to be practical under high-dimensional, sparse sequences involving thousands of event types—a common trait in real-world data. We propose OSCAR, a novel one-shot causal autoregressive discovery method that identifies the Markov Boundaries of each label directly from a single sequence of events. By leveraging two pretrained Transformers as density estimators, OSCAR estimates the conditional mutual information between the current event and future labels given the past sequence, enabling for the first time efficient parallelised causal discovery on GPUs. On a real-world vehicle dataset with 29,100 event types and 474 labels, OSCAR successfully recovers meaningful causal structures where classical algorithms fail to scale, demonstrating a practical path toward interpretable and efficient causal reasoning in complex sequential domains.

## 1 INTRODUCTION

Causal discovery in event sequences is a central problem across domains such as cybersecurity Manocchio et al. (2024), healthcare Rasmy et al. (2020); He et al. (2022), flight operations Luo et al. (2021) or vehicle defects Pirasteh et al. (2019). These sequences, composed of discrete asynchronous events, are increasingly available at scale– yet remain challenging to interpret beyond associations. Understanding *why* specific events lead to particular outcomes is vital for effective diagnosis, prediction and overall decision making Liu et al. (2025); Qiao et al. (2023).

Transformers have significantly advanced sequence modelling by capturing complex data distribution through self-attention and autoregressive factorisation Vaswani et al. (2017); Radford et al. (2018); Touvron et al. (2023). While they excel at next-token prediction, recent works explore their use for causal discovery by interpreting attention scores Nauta et al. (2019); Alonso et al. (2024); Rohekar et al. (2023) or using Transformers as density estimators for causal inference Im et al. (2024); Moghimifar et al. (2020).

However, the majority of existing causal discovery methods, such as constraint-based or Granger-style approaches, remain computationally intractable in high-dimensional event sequences involving thousands of event types, due to the number of CI-tests involved. Additionally, their goal is often to recover a global graph, which is rarely interpretable or actionable in real-world environments.

In contrast, practitioners frequently reason about causality *within individual unknown sequences*. For instance, "what series of events captured by diagnostics led to this vehicle failure" or "what symptoms led to this disease". Here, an event sequence consists of a list of discrete events $x_i$ recorded asynchronously over time, while labels $\boldsymbol{y}$ summarise outcomes associated with the full sequence (e.g, a diagnosed defect or condition).

We aim to solve this setting in a one-shot manner: given only a single unknown sequence of observed events, we directly infer the causal structure explaining its outcomes, without needing multiple repetitions or large aggregated datasets. Specifically, we seek to extract, for each label, the minimal set of causal events—its Markov Boundary.

In this work, we address this gap by introducing OSCAR: the first One-Shot multi-label Causal AutoRegressive discovery method. It leverages two Transformers as density estimators to estimate conditional mutual information Cover (1999) using natural language processing sampling techniques Holtzman et al. (2020). In this manner, instead of learning global structure, OSCAR extracts a compact interpretable subgraph with causal indicators between events and labels, providing better explainability. Unlike traditional causal discovery methods that suffer label cardinality-dependent time complexity Li et al. (2016); Yu et al. (2020); Hasan et al. (2023); Gong et al. (2024), OSCAR supports causal discovery across thousands of event types and hundreds of labels. Thanks to its fully parallelised structure, it provides sequence-specific explainability in a matter of minutes for thousands of sequences and reuse existing pretrained sequence models as backbones, making it easily applicable in production.

We validate our approach on a real-world vehicular dataset comprising 29,100 event types as diagnosis trouble codes and 474 labels as error patterns (EPs) representing vehicle defects Math et al. (2025). By setting the known EPs rules as ground truth Markov Boundaries, we benchmark OSCAR against standard well-established causal discovery baselines and demonstrate its practical superiority in accuracy and scalability. To the best of our knowledge, this is the first method that solve efficiently multi-label causal discovery for high-dimensional event sequences.

The contributions of this paper are as follows: 1) We introduce OSCAR, the first one-shot multi-label causal discovery method that identifies Markov Boundaries of labels from high-dimensional event sequences in parallel using Transformer-based as density estimators. 2) We provide theoretical guarantees under several assumptions, showing that when using an estimation of conditional mutual information (CMI), we can identify the correct Markov Boundaries of each label from a single event sequences. 3) We empirically validate OSCAR on a large-scale vehicular dataset, demonstrating its scalability and practical superiority over traditional causal discovery baselines.

## 2 RELATED WORK

**Event Sequence Modelling.** Event sequences are typically represented as a series of time-stamped discrete events $S = \{(t_1, x_1), \ldots, (t_L, x_L)\}$ where $0 \le t_1 < \ldots \le t_L$ denotes the time of occurrence of event type $x_i \in \mathbb{X}$ drawn from a finite vocabulary $\mathbb{X}$. In multi-label settings, a binary label vector $\boldsymbol{y} \in \{0, 1\}^{|\mathbb{Y}|}$ is attached to $S$ and denotes the presence of multiple outcome labels drawn from $\mathbb{Y}$ occuring at final time step $t_L$. Forming a multi-labeled sequence $S_l = (S, (\boldsymbol{y}_L, t_L))$.

Event sequence modelling has been widely applied to predictive tasks. For instance, in the automotive domain, Diagnostic Trouble Codes (DTCs) Pirasteh et al. (2019) are logged asynchronously over time and used to infer failures or error patterns Math et al. (2025). In healthcare, electronic health records encode temporal sequences of symptoms, test results, and treatments that are predictive of downstream diagnosis Rasmy et al. (2020); Labach et al. (2023); He et al. (2022). A common modelling strategy Lafferty et al. (2001); McCallum et al. (2000) separates such event types $\mathbb{X}$ from labels $\mathbb{Y}$, thus it becomes easier to perform prediction tasks due to the difference in cardinality between them.

Transformers Vaswani et al. (2017) have emerged as the dominant architecture for sequence modelling, thanks to their ability to model long-range dependencies through self-attention. Recent work has leveraged Transformers in high-dimensional event spaces for next-event and label prediction. Notably Math et al. (2025) proposed a dual Transformer architecture where one model predicts the next event type (DTC), and the other predicts the label occurrence (e.g, error pattern). Through this paper, we build on this dual architecture and extend it beyond predictive modelling toward causal discovery.

**Neural Autoregressive Density Estimation.** Neural autoregressive models were initially introduced for density estimation via chain-rule factorisation of the joint distribution Bengio & Bengio (1999), later extended through recurrent architectures Cho et al. (2014); Hochreiter & Schmidhuber (1997) and Transformers Vaswani et al. (2017). These models are trained using next-token prediction, by minimising the negative log-likelihood of observing sequences $X = (x_1, \ldots, x_L)$. The joint probability can be expressed as:

$$P(X) = \prod_{i=1}^{L} P(x_i \mid x_1, \ldots, x_{i-1}). \tag{1}$$

Recent work has explored autoregressive models as tools for causal inference. For example, Garrido et al. (2021) leverages density estimators to simulate interventions and compute average treatment effects. Im et al. (2024) shows that autoregressive language models can approximate sequential Bayesian networks (Fig .1), treating the model itself as a statistical engine for causal inference. These findings motivate our use of pretrained Transformers to estimate conditional mutual information (CMI) Cover (1999) between events and labels.

In temporal data, Granger (1969) causality is commonly employed to assess pairwise dependencies Xu et al. (2016); Qiao et al. (2023), based on the assumption that causes precede effects and should improve the predictability of the effect. Recently, Han et al. (2025) proposed a Granger-inspired causal discovery framework in multivariate time series using an encoder-decoder architecture. Specifically, we repurposed these models as neural autoregressive density estimators (NADEs) for both the events and labels, allowing us to quickly estimate the conditional probabilities of the next event $x_i$ and labels $y$ given past events $(x_1, \cdots, x_{i-1})$.

**Transformers as Causal Learners.** Transformer-based models have gained growing attention in the causal discovery literature. Nichani et al. (2024) showed that when trained on sequences generated from in-context Markov chains, they can implicitly learn latent causal graphs, where attention weights align with the adjacency matrix of the true causal structure. For sequential data, Rohekar et al. (2023) analyses self-attention under the assumption that data is generated by a linear-Gaussian structural causal model (SCM) Spirtes et al. (2001). They relate the covariance of endogenous variables to attention scores and apply conditional independence (CI) tests to the final layer's outputs to recover a partial ancestral graph. Our work builds on this idea by leveraging Transformers but focuses on multi-label event sequences. Although they refer to it as *zero-shot*, we found that *one-shot* is more explicit since it requires a single sequence from unseen data of the same domain to infer a graph.

**Multi-label Causal Discovery.** Multi-label causal discovery seeks to identify the Markov Boundary (**MB**) of each label—its minimal set of parents, children, and spouses—such that the label is conditionally independent of all other variables given its **MB** Tsamardinos & Aliferis (2003). This boundary serves as an optimal feature set for tasks like explainable modelling and feature selection, under the faithfulness assumption.

While classical constraint-based algorithms have shown success on low-dimensional tabular data Spirtes & Glymour (1991); Yu et al. (2020), their application to event sequences with multi-label outputs remains challenging due to: (1) *dimensionality*—thousands of event types increase the number of potential interactions combinatorially; (2) *sparsity*—multi-hot encodings often underrepresent rare but important events; (3) *temporal dependencies*—causal effects can occur with varying delays; and (4) *distributional assumptions*—such as linearity or Gaussian noise, which rarely hold in real-world sequences.

Some recent works attempt to address these challenges. CASCADE Cüppers et al. (2024) recovers DAGs from temporal event data under a Poisson process assumption but is limited to smaller event spaces ($\sim 200$ types). Qiao et al. (2023) explore Granger causality under low-resolution temporal data using Hawkes processes Hawkes (1971) and show gains in F1 across time granularities, though their setup also assumes relatively small event vocabularies. However, rather than learning the full joint causal graph, which is known to be NP-hard Chickering (1996)—we focus on recovering local causal structure (LCS) Yu et al. (2020): discovering minimal subgraphs from inputs to labels within a single sequence. This formulation makes the problem tractable in high dimensions and better suited for real-world production scenarios.

Hence, contrary to event-to-event causal learning, multi-label causal discovery remains unexplored in event sequences Gong et al. (2024); Hasan et al. (2023), yet it's potential applications are enormous across various domains. Making OSCAR a novel method to explain high-dimensional labeled event sequences. We focus on causal discovery only and not event sequence modeling as in Math et al. (2025).

## 3 NOTATIONS AND DEFINITIONS

We use capital letters (e.g., $X$) to denote random variables, $P(X)$ the probability distribution of $X$, $P(X = x) = p(x)$ the probability of the realisation $x$ for the random variable $X$, and bold capital letters (e.g., $\boldsymbol{X}$) for sets of variables. Let $\boldsymbol{U}$ denote the set of all (discrete) random variables. We

define the event set $\boldsymbol{X} = \{X_1, \ldots, X_n\} \subset \boldsymbol{U}$, and the label set $\boldsymbol{Y} = \{Y_1, \ldots, Y_n\} \subset \boldsymbol{U}$. When explicitly said, event $X_i^{(t_i)}$ represent the occurrence of $X_i$ at step $i$ and time $t_i$. Similarly for $Y_{i+1}^{(t_{i+1})}$.

**Definition 1** (Bayesian Network). *Pearl (1988) Let $P$ denote the joint distribution over a variable set $\boldsymbol{U}$ of a directed acyclic graph (DAG) $\mathcal{G}$. The triplet $< \boldsymbol{U}, \mathcal{G}, P >$ constitutes a BN if the triplet $< \boldsymbol{U}, \mathcal{G}, P >$ satisfies the Markov condition: every random variable is independent of its non-descendant variables given its parents in $\mathcal{G}$. Each node $X_i \in \boldsymbol{U}$ represents a random variable. The directed edge $(X_i \to X_j)$ encodes a probabilistic dependence. The joint probability distribution can be factorized $P(X_1, \cdots, X_n) = \prod_{i=1}^n P(X_i | X_1, \cdots, X_{i-1})$. If a variable does not depend on all of its predecessors, we can write: $P(X_i | X_1, \cdots, X_{i-1}) = P(X_i | par(X_i))$ with 'par' the parents of node $X_i$ such that: $par(X_i) = \{X_1, \cdots, X_{i-1}\}$.*

**Definition 2** (Faithfulness). *Spirtes et al. (2001). Given a BN $< \boldsymbol{U}, \mathcal{G}, P >$, $\mathcal{G}$ is faithful to $P$ if and only if every conditional independence present in $P$ is entailed by $\mathcal{G}$ and the Markov condition holds. $P$ is faithful if and only if there exist a DAG $\mathcal{G}$ such that $\mathcal{G}$ is faithful to $P$.*

**Definition 3** (Markov Boundary). *Tsamardinos & Aliferis (2003). In a faithful BN $< \boldsymbol{U}, \mathcal{G}, P >$, for a set of variables $\boldsymbol{Z} \subset \boldsymbol{U}$ and label $Y \in \boldsymbol{U}$, if all other variables $X \in \{\boldsymbol{X} - \boldsymbol{Z}\}$ are independent of $Y$ conditioned on $\boldsymbol{Z}$, and any proper subset of $\boldsymbol{Z}$ do not satisfy the condition, then $\boldsymbol{Z}$ is the Markov Boundary of $Y$: $\boldsymbol{MB}(Y)$.*

**Definition 4** (Conditional Independence). *Variables $X$ and $Y$ are said to be conditionally independent given a variable set $\boldsymbol{Z}$, if $P(X, Y | \boldsymbol{Z}) = P(X | \boldsymbol{Z})P(Y | \boldsymbol{Z})$, denoted as $X \perp Y | \boldsymbol{Z}$. Inversely, $X \not\perp Y | \boldsymbol{Z}$ denotes the conditional dependence. Using the conditional mutual information Cover (1999) to measure the independence relationship, this implies that $I(X, Y | \boldsymbol{Z}) = 0 \Leftrightarrow X \perp Y | \boldsymbol{Z}$.*

**Auto-regressive Event Sequence Models.** We reuse the two architectures introduced by Math et al. (2025) to perform next event prediction (*CarFormer* as $\text{Tf}_x$) and next labels (*EPredictor* as $\text{Tf}_y$). These two autoregressive Transformers model the conditional probability distribution of the next events and labels conditioned on the past sequence of observed events $\boldsymbol{Z} = (x_1, \cdots, x_{i-1}) = S_{<i}$, the predictive distributions are:

$$\text{Tf}_x(S_{<i}) = \textit{Softmax}(\boldsymbol{h}_{i-1}^x) \triangleq P_{\theta_x}(X_i | \boldsymbol{Z}) \tag{2}$$

$$\text{Tf}_y(S_{\leq i}) = \textit{Sigmoid}(\boldsymbol{h}_i^y) \triangleq P_{\theta_y}(Y | X_i, \boldsymbol{Z}) \tag{3}$$

Here, $\boldsymbol{h}_{i-1}^x, \boldsymbol{h}_i^y \in \mathbb{R}^d$ are the logits produced by the two Transformer heads of $\text{Tf}_x$ and $\text{Tf}_y$ parametrized by $\theta_x, \theta_y$. The majority of $\text{Tf}_x$ (expect the heads) serves as a backbone for $\text{Tf}_y$.

## 4 METHODOLOGY

Working with causal structure learning from observed data requires several assumptions, notably the causal Markov assumptions Pearl (1988) states that a variable is conditionally independent of its non-descendants given its parents. An extended discussion on the impact of assumptions is provided later in Appendix E, and proofs in Appendix B. We assume the following:

**Assumption 1** (Temporal Precedence). *Given a perfectly recorded sequence of events $((x_1, t_1), \cdots, (x_L, t_L))$ with labels $(\boldsymbol{y}_L, t_L)$ and monotonically increasing time of occurrence $0 \leq t_1 \leq \cdots \leq t_L$, an event $x_i$ is allowed to influence any subsequent event $x_j$ such that $t_i \leq t_j$ and $i < j$. Formally, the graph $\mathcal{G} = (\boldsymbol{U}, \boldsymbol{E})$, $(x_i, x_j) \in \boldsymbol{E} \implies t_i \leq t_j$ and step $i < j$*

Figure 1: An example of a causal graph extracted from a multi-label event sequence where $\text{MB}_1$ represents the Markov Boundary of $Y_1$ and $\text{MB}_2$ the Markov Boundary of $Y_2$.

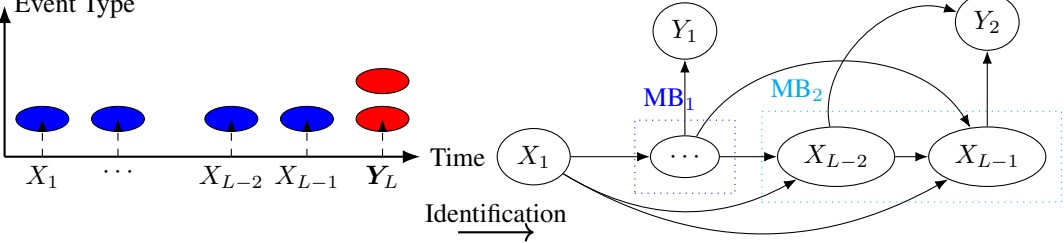

It allows us to remove ambiguity in causal directionality and orient the BN edges in Fig .1.

**Assumption 2** (Causal Sufficiency for Labels). *All relevant variables are observed, and there are no hidden confounders affecting the labels.*

**Assumption 3** (Oracle Models). *We assume that two autoregressive Transformer models, $Tf_x$ and $Tf_y$, are trained via maximum likelihood on a dataset of multi-labeled event sequences $D_n = \{S_l^1, \cdots, S_l^n\} \subset \mathbb{S}$, and can perfectly approximate the true conditional distributions of events and labels:*

$$P(X_i|Pa(X_i)) = P_{\theta_x}(X_i|Pa(X_i)) = Tf_x(S_{<i}), \ P(Y_j|Pa(Y_j)) = P_{\theta_y}(Y_j|Pa(Y_j)) = Tf_y(S_{\leq i}) \tag{4}$$

**Assumption 4** (Bounded Lagged Effects). *Once we observed events up to timestamp $t_i$ and step $i$ as $\mathbf{Z}_{\leq t_i} = ((x_1, t_1), \cdots, (x_i, t_i))$, any future lagged copy of event $X_i^{(t_i+\tau)}$ is independent of $Y_j$ conditioned on $\mathbf{Z}_{\leq t_i}$:*

$$Y_j \perp X_i^{(t_i+\tau)}|\mathbf{Z}_{\leq t_i}$$

*Where $\tau = t_{i+1} - t_i$ is a finite bound on the allowed time delay for causal influence.*

In other words, we allow the causal influence of event $X_i$ on $Y_j$ until the next event $X_{i+1}$ is observed. We note that for data with strong lagged effects (e.g., financial transactions), this might not hold well, but relevant for log-based and error code-based data.

**Lemma 1** (Identifiability of $\mathbb{G}$). *Assuming the faithfulness condition holds for the true causal graph $\mathbb{G}$. Let $Tf_x$ and $Tf_y$ be oracle models that model the true conditional distributions of events and labels, respectively. The joint distribution $P_{\theta_x, \theta_y}$ can then be constructed, and any conditional independence detected from the distributions estimated by $Tf_x$ and $Tf_y$ corresponds to a conditional independence in $\mathbb{G}$:*

$$X_i \perp_{\theta_x, \theta_y} Y_j \mid \mathbf{Z} \quad \implies \quad X_i \perp_{\mathbb{G}} Y_j \mid \mathbf{Z}.$$

*Where $\perp_{\theta_x, \theta_y}$ denotes the independence entailed by the joint probability $P_{\theta_x, \theta_y}$.*

**Lemma 2** (Markov Boundary Equivalence). *In a multi-label event sequence $S_l$ and under the temporal precedence assumption A1, the Markov Boundary of each label $Y_j$ is only its parents such that $\forall X \in \{\mathbf{U} - Pa(Y_j)\}, X \perp Y_j|Pa(Y_j) \Leftrightarrow MB(Y_j) = Pa(Y_j)$.*

**Theorem 1** (Markov Boundary Identification in Event Sequences). *If $S_l^k$ a multi-labeled sequence drawn from a dataset $D_n = \{S_l^1, \cdots, S_l^n\} \subset \mathbb{S}$ where two Oracle Models $Tf_x$ and $Tf_y$ were trained on, then under causal sufficiency (A2), bounded lagged effects (A4) and temporal precedence (A1), the Markov Boundary of each label $Y_j$ in the causal graph $\mathbb{G}$ can be identified using conditional mutual information for CI-testing.*

We prove Theorem 1 in Appendix B.3 by induction. Such that under the previous assumptions, we can correctly sequentially recover the Markov Boundary of our labels in the associated BN (Def 1).

Figure 2: The overview of OSCAR: One-Shot multi-label Causal AutoRegressive discovery. $d$ denotes the hidden dimension, $L$ the sequence length, $MB_1, MB_2$ the Markov Boundary of $Y_1, Y_2$ respectively. All blue and green areas are parallelized on GPUs.

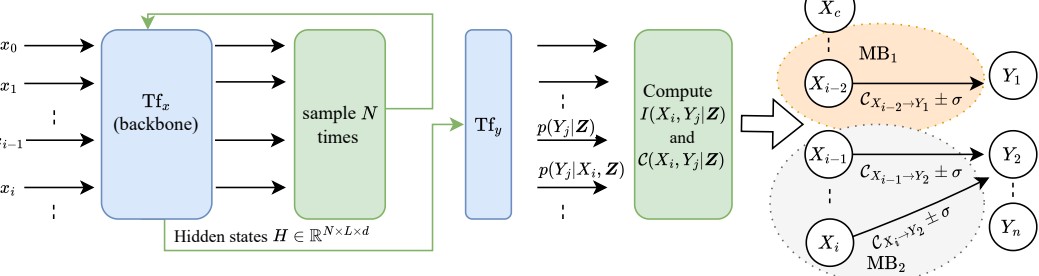

## 4.1 CONDITIONAL MUTUAL INFORMATION ESTIMATION VIA AUTOREGRESSIVE MODELS

OSCAR works like a constraint-based causal discovery algorithm where the conditioning set of nodes $\boldsymbol{Z}$ increases over time. Event apparitions are modelled using a sequential BN (Fig .1). Specifically, we would like to access how much additional information event $X_i$ occurring at step $i$ provides about label $Y_j$ when we already know the past sequence of events $\boldsymbol{Z} = S_{<i}$. We essentially try to answer if:

$$P(Y_j|X_i, \boldsymbol{Z}) = P(Y_j|\boldsymbol{Z}) \Leftrightarrow D_{KL}(P(Y_j|X_i, \boldsymbol{Z})\|P(Y_j|\boldsymbol{Z})) = 0$$

where $D_{KL}$ denotes the *Kullback-Leibler divergence* Cover (1999). The distributional difference between the conditionals $P(Y_j|X_i, \boldsymbol{Z}), P(Y_j|\boldsymbol{Z})$ is akin to Information Gain $I_G$ Quinlan (1986) conditioned on past events:

$$I_G(x_i, Y_j|z_i) \triangleq D_{KL}(P(Y_j|X_i = x_i, \boldsymbol{Z} = z_i))\|P(Y_j|Z = z_i)) \tag{5}$$

Which is equals to the difference between the conditional entropies Cover (1999); Quinlan (1986) denoted as $H$:

$$I_G(Y_j, x_i|z_i) = H(Y_j|z_i) - H(Y_j|x_i, z_i) \tag{6}$$

More generally, we can use the CMI to assess conditional independence (Def 4) which is simply the expected value of the information gain $I_G(Y_j, x_i|z_i)$ such as:

$$I(Y_j, X_i|\boldsymbol{Z}) \triangleq H(Y_j|\boldsymbol{Z}) - H(Y_j|\boldsymbol{Z}, X_i) = \mathbb{E}_{x_i, z_i}[I_G(Y_j, X_i = x_i|\boldsymbol{Z} = z_i)]) \tag{7}$$

It can be interpreted as the expected value over all possible context $\boldsymbol{Z}$ of the deviation from independence of $X_i, Y_j$ in this context. To approximate equation 7, a naive Monte Carlo Doucet et al. (2001) approximation is performed where we draw $N$ random variations of the conditioning set $z^{(l)} = \{x_0^{(l)}, \ldots, x_{i-1}^{(l)}\}$, denoting the $l$-th sampled particle:

$$\hat{I}_N(X_{i+1}, X_i \mid \boldsymbol{Z}) = \frac{1}{N} \sum_{l=1}^{N} I_G(X_{i+1}, X_i \mid \boldsymbol{Z} = z^{(l)}) \tag{8}$$

This estimator is unbiased because the contexts $z^{(l)}$ are sampled directly from $\mathrm{Tf}_x$ using a proposal $Q$ with the same support as $P(\boldsymbol{Z})$. Since $I_G(X_{i+1}, X_i \mid \boldsymbol{Z} = z)$ is a difference between conditional entropies (equation 6), it is thus bounded uniformly Cover (1999) by the log of supports such as:

$$0 < I_G(X_{i+1}, X_i \mid \boldsymbol{Z} = z^{(l)}) = H(X_{i+1}|z^{(l)}) - H(X_{i+1}|x_i, z^{(l)})) \leq H(X_{i+1}) \leq \log|\mathbb{X}|$$

Thus the posterior variance of $f_i = I_G(X_{i+1}, X_i \mid \boldsymbol{Z} = z^{(l)})$ satisfies $\sigma_{f_i}^2 \triangleq \mathbb{E}_{p(z)}[f_i^2(p(z)] - I^2(f_i) < +\infty$ Doucet et al. (2001) then the variance of $\hat{I}_N(f_i))$ is equal to $var(\hat{I}_N(f_i)) = \frac{\sigma_{f_t}^2}{N}$ and from the strong law of large numbers:

$$\hat{I}_N \xrightarrow[N \to +\infty]{\text{a.s.}} \mathbb{E}_z[I_G(X_{i+1}, X_i \mid \boldsymbol{Z} = z)] \triangleq I(f_i). \tag{9}$$

An ablation of different proposal $Q$ is presented in Appendix D.2. where we also study the effect of $N$ on classification and computational cost. Empirically, we found that combining top-$k = 35$ (randomly taking the $k$ most probable tokens for each step) with nucleus sampling Holtzman et al. (2020) ($p = 0.9$) and $N = 68$ provided the best trade-off between performance and efficiency.

In practice a label-specific threshold $\theta_j \approx 0$ is applied to equation 8 to identify conditional independence:

$$Y_j \not\perp X_i \mid \boldsymbol{Z} \quad \Leftrightarrow \quad I(Y_j, X_i \mid \boldsymbol{Z}) > \theta_j \approx 0 \tag{10}$$

$\theta_j$ is dynamically computed for each label based on the mean and standard deviation of the CMI values across the sequence such that: $\theta_j = \mu_{Y_j} + k \cdot \sigma_{Y_j}$, where $k$ controls the confidence interval. We analyze the effect of $k$ in Fig. 5.

To ensure stable conditional entropy estimates and reliable predictions from $\mathrm{Tf}_y$, the CMI is computed after observing $c$ events (*context*). This design choice also enables out-of-the-box parallelisation.

By sampling $N$ variations of the prefix sequence $S_{\leq c}$, the CMI is independently computed across positions $i \in [c, L]$. One caveat is the phenomenon of entropy saturation Shannon (1951), whereby the conditional entropy $H(Y_j \mid \boldsymbol{Z}_i)$ diminishes as $\boldsymbol{Z}_i = S_{\leq i}$ grows longer:

$$H(Y_j \mid X_{i+1}, \boldsymbol{Z}_i) \leq H(Y_j \mid X_i, \boldsymbol{Z}_{i-1}).$$

In other words, once a sufficiently informative context is observed, future uncertainty becomes minimal. Therefore, context $c$ and sequence length $L$ must be carefully selected to balance informativeness and computational efficiency. In our case, we set $c = 15, L = 128$ for our experiments. An ablation on $c$ and the quality of the NADEs can be found in the Appendix D.1, as well as an extended discussion on the assumptions in E.

### 4.1.1 COMPUTATION

A key advantage of our approach is its scalability. Unlike traditional methods whose complexity depends on the event and label cardinality $|\mathbb{X}|$ and $|\mathbb{Y}|$ Li et al. (2016), our method is agnostic to both. CMI estimations are independently performed for all positions $i \in [c, L]$, with the sampling pushed into the batch dimension and results averaged across labels, leading to BS $\times N \times L$ CI-tests per batch $D = \{S_l^0, \dots, S_l^n\}$.

Consequently, time complexity transitions from $\mathcal{O}(\text{BS} \times N \times L)$ to $\mathcal{O}(1)$ per batch due to GPU parallelism. The complexity is bounded by the Transformers' inference part, where it scales quadratically with the sequence length $\mathcal{O}(L^2)$ if one uses vanilla self-attention Vaswani et al. (2017). The implementation of OSCAR in *Pytorch* Paszke et al. (2019) is provided in Appendix G. It can be easily decomposed into several steps such as:

```
1 logits_x = tfx(**batch)['prediction_logits']
2 x_hat = F.softmax(logits_x, dim=-1)
3 sampled = topk_p_sampling(batch['input_ids'], x_hat, c=c, n=N)
```

Listing 1: Step 1: Next-event prediction and sampling.

The event Transformer `tfx` produces logits over next event types. We apply top-$k$/nucleus sampling to expand the batch into $N$ candidates in parallel. Only the first $c$ events are sampled.

```
1 out_y = tfy(input_ids=sampled.reshape(-1,L), attention_mask=
       attention_mask.repeat(N, 1))
2 prob_y = torch.sigmoid(out_y['logits']).reshape(bs, N, L-c, -1)
3 prob_y = torch.clamp(prob_y, eps, 1-eps)
```

Listing 2: Step 2: Next-label prediction.

The label Transformer `tfy` evaluates all samples in one forward pass starting from $c$, yielding conditional probabilities $P(Y_j|\boldsymbol{Z})$ and $P(Y_j|X_i, \boldsymbol{Z})$. We then calculate the binary $D_{KL}$:

```
1 y_z, y_zx = prob_y[...,:-1,:], prob_y[...,1:,:]
2 cmi = torch.mean(
3     y_zx*torch.log(y_zx/y_z) +
4     (1-y_zx)*torch.log((1-y_zx)/(1-y_z)),
5     dim=1
6 )   # (bs, L, |Y|)
```

Listing 3: Step 3: Conditional mutual information.

Conditional mutual information is averaged across the sampling dimension, producing a compact $(\text{bs}, L - c, |\mathbb{Y}|)$ tensor:

```
1 mu, std = cmi.mean(dim=1), cmi.std(dim=1)
2 mask = cmi >= (mu + k*std).unsqueeze(1)
```

Listing 4: Step 4: Dynamic thresholding.

Finally, dynamic per-label thresholds identify causal events based on their value across the sequence length dimension.

## 4.2 CAUSAL INDICATOR

While deterministic DAGs reveal structural dependencies, they often obscure the *magnitude* and *direction* of influence between variables. In many settings, a small subset of causal events may exert disproportionate influence on the probability of a label. Moreover, causal relationships can be either *excitatory* or *inhibitory*—that is, the presence of a cause may either increase or decrease the likelihood of its effect.

For instance, if $P(Y_j \mid X_i, \boldsymbol{Z}) < P(Y_j \mid \boldsymbol{Z})$ then $X_i$ negatively influences $Y_j$, yet still constitutes a valid causal relationship Pearl (2009). Without quantifying the effect direction and strength, such cases may mislead the operator. Given that we can estimate both conditionals $P(Y_j \mid X_i, \boldsymbol{Z})$ and $P(Y_j \mid \boldsymbol{Z})$, we define the *causal indicator* $\mathcal{C} \in [-1, 1]$ between an event $X_i$ and a label $Y_j$ under context $\boldsymbol{Z}$ that we assume fixed for every measurement Fitelson & Hitchcock (2010):

$$\mathcal{C}(Y_j, X_i; \boldsymbol{Z}) := \mathbb{E}_Z[P(Y_j \mid X_i, Z) - P(Y_j \mid Z)]$$

following the measure proposed by Eells (1991). Here, $\mathcal{C} > 0$ indicates a positive influence and $\mathcal{C} < 0$ reflects an inhibitory effect. While several metrics for causal strength exist—including Causal Power Cheng (1997) and Good's measure Fitelson & Hitchcock (2010), we adopt Eells' measure for its simplicity of interpretation. An operator can easily read it and get a sense of the raise in likelihood of the label $Y_j$ We employ the term causal *indicator* to separate from causal strength measures, which, if using this formulation, can be problematic as pointed out by Janzing et al. (2012). Ours serves more as an indication than a strength, which is here the conditional mutual information.

$\mathcal{C}$ is computed using the same Monte-Carlo simulation as in equation 8 by averaging over all sampled contexts. We compute mean and standard deviations over contexts to provide uncertainty estimates.

## 5 EMPIRICAL EVALUATION

**Settings.** We used a $g4dn.12xlarge$ instance from AWS Sagemaker to run comparisons. It contains 48 vCPUs and 4 NVIDIA T4 GPUs. During inference, we used fp16 for $\text{Tf}_y$ and fp32 for $\text{Tf}_x$. We used a combination of F1-Score, Precision, and Recall with different averaging Zhang & Zhou (2014) (Appendix C.1) to perform the comparisons. The code for OSCAR, $\text{Tf}_x, \text{Tf}_y$ and the evaluation are provided anonymously [1] as well as the anonymised version of the dataset for reproducibility purposes.

**Vehicle Event Sequences Dataset.** We evaluated our method on a real-world vehicular test set of $n = 300,000$ sequences. It contains $|\mathbb{Y}| = 474$ different error patterns and about $|\mathbb{X}| = 29,100$ different DTCs forming sequences of $\approx 150 \pm 90$ events. We used 105m backbones as $\text{Tf}_x, \text{Tf}_y$ Math et al. (2025). The two NADEs didn't see the test set during training. The two NADEs didn't see the test set during training. The error patterns are manually defined by domain experts as boolean rules between DTCs. For instance, in equation 11, DTCs $x_1$ is a cause of the error pattern $y_1$. We set the elements of this rule as the correct Markov Boundary for each label $y_j$ in the tested sequences. It is important to note that rules are subject to changes over time by domain experts, making it more difficult to extract the true **MB**. Moreover, there is about 12% missing ground truth **MB** rules for certain $Y_j$.

Figure 3: Example of an error pattern $(y_1)$ boolean definition based on diagnosis trouble codes $(x_i)$

$$y_1 = x_1 \ \& \ (x_5 \mid x_8) \ \& \ (x_{18} \mid x_{12}) \ \& \ x_3 \ \& \ (!x_{10} \mid !x_{20}) \tag{11}$$

**Comparisons.** Although no existing method directly targets one-shot multi-label causal discovery Gong et al. (2024), we benchmark OSCAR against local structure learning (LSL) algorithms that estimate global Markov Boundaries. This includes established approaches such as CMB Gao & Ji (2015), MB-by-MB Wang et al. (2014), PCD-by-PCD Yin et al. (2008), IAMB Tsamardinos et al. (2003) from the *PyCausalFS* package Yu et al. (2020), as well as the more recent, state-of-the-art

---

[1] https://tinyurl.com/oscar-iclr-2026

MI-MCF Ma et al. (2025). 9 random folds of the test data were created and converted into a multi-one-hot data-frame where one row represents one sequence and each column represents an event type or label $(\mathbb{X}, \mathbb{Y})$. We set the target nodes as the labels with *PyCausalFS*.

**Performances.** We first drew $n = 50,000$ random sequences from our dataset and performed comparisons (Table 1). We found out that even under this reduced setup, LSL algorithms failed to compute the Markov Boundaries within a 3 days timeout, far exceeding practical limits for deployment. OSCAR on the other hand, shows robust classification over a large amount of events $(29,100)$, especially $55\%$ precision, in a matter of minutes. This behavior highlights the current infeasibility of multi-label causal discovery in high-dimensional event sequences. This positions OSCAR as a more feasible approach for large-scale causal per-sequence causal reasoning in production environments.

To enable at least partial comparison, we further sub-sampled to $n = 500$ sequences (Table 2) to enable a faster computation. However, there is about the same number of labels in the test set for $n = 500$ samples. Resulting to a poorly number of CI-tests for the baselines. As a result, LSL algorithms output empty **MB** sets after multiple hours. Especially MI-MCF with even 500 samples suffers from its expensive CMI testing. Thus, traditional algorithms suffer from having either too much samples and taking days to compute or too less data to even function. This positions OSCAR as a more feasible approach for large-scale, multi-label causal discovery in event sequences.

Table 1: Comparisons of **MB** retrieval with $n = 50,000$ samples, $|\mathbb{X}| = 29,100, |\mathbb{Y}| = 474$ averaged over $6-$folds. Classification metrics averaging is 'weighted' and shown as one-shot for OSCAR. The symbol '-' indicates that the algorithm didn't output the **MBs** under 3 days. Metrics are given in $\%$.

| Algorithm | Precision↑ | Recall↑ | F1↑ | Running Time (min)↓ |
|---|---|---|---|---|
| IAMB | - | - | - | $> 4320$ |
| CMB | - | - | - | $> 4320$ |
| MB-by-MB | - | - | - | $> 4320$ |
| PCDbyPCD | - | - | - | $> 4320$ |
| MI-MCF | - | - | - | $> 4320$ |
| OSCAR | $\mathbf{55.26 \pm 1.42}$ | $\mathbf{31.37 \pm 0.82}$ | $\mathbf{40.02 \pm 1.03}$ | $\mathbf{11.7}$ |

Table 2: Comparisons of **MB** retrieval with $n = 500$ samples over $9-$folds.

| Algorithm | Precision↑ | Recall↑ | F1 ↑ | Running Time (min)↓ |
|---|---|---|---|---|
| IAMB | $0.0 \pm 0.0$ | $0.0 \pm 0.0$ | $0.0 \pm 0.0$ | $129.4$ |
| CMB | $0.0 \pm 0.0$ | $0.0 \pm 0.0$ | $0.0 \pm 0.0$ | $128.7$ |
| PCDbyPCD | $0.0 \pm 0.0$ | $0.0 \pm 0.0$ | $0.0 \pm 0.0$ | $129.1$ |
| MB-by-MB | $0.0 \pm 0.0$ | $0.0 \pm 0.0$ | $0.0 \pm 0.0$ | $140.3$ |
| MI-MCF | $0.0 \pm 0.0$ | $0.0 \pm 0.0$ | $0.0 \pm 0.0$ | $> 1440$ |
| OSCAR | $\mathbf{54.78 \pm 2.91}$ | $\mathbf{30.39 \pm 2.39}$ | $\mathbf{39.92 \pm 2.25}$ | $\mathbf{0.14}$ |

We exemplify the explainability provided by our method for the task of explaining error patterns happening to a vehicle (Fig .10). A concrete use case for OSCAR in this context would be to refine or create new error pattern rules based on OSCAR output predictions, such as non-common causal variables Wu et al. (2020) between labels (*Camera Error* node), leading to a better automation of quality processes. More examples are given in the Appendix F.1.

## 6 CONCLUSION

We presented OSCAR, the first scalable one-shot causal discovery method for high-dimensional multi-labeled event sequences. It succeeded in uncovering causal structures on a real-world dataset in an order of minutes, while classical baselines failed under the strain of dimensionality. Beyond local structure learning, OSCAR quantifies causal strengths, offering more actionable insights in contrast to deterministic DAGs.

OSCAR marks a decisive step towards making causality practical and efficient on GPUs for complex, real-world high-dimensional sequential data.

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

## A  APPENDIX

## B  PROOFS

We provide proofs for the results described in Section 4

### B.1  PROOF OF LEMMA 1

*Proof.* We assume that the data is generated by the associated causal graph $\mathcal{G}$ following the sequential BN from a multi-labeled sequence $S$. And that the faithfulness assumption holds Pearl (1988), meaning that all conditional independencies in the observational data are implied by the true causal graph $\mathcal{G}$.

Given that the Oracle models $\text{Tf}_x$ and $\text{Tf}_y$ are trained to perfectly approximate the true conditional distributions, for any variable $U_i$ in the graph, we have:

$$P(U_i|\text{Pa}(U_i)) = \begin{cases} P(Y_j|\text{Pa}(Y_j)) = P_{\theta_y}(Y_j|\text{Pa}(Y_j)), & \text{if } U_i \in \boldsymbol{Y} \\ P(X_i|\text{Pa}(X_i)) = P_{\theta_x}(X_i|\text{Pa}(X_i)), & \text{otherwise.} \end{cases}$$

By the faithfulness assumption Pearl (1988), if the conditional independencies hold in the data, they must also hold in the causal graph $\mathcal{G}$:

$$X_i \perp Y_j|\boldsymbol{Z} \implies X_i \perp_{\mathcal{G}} Y_j|\boldsymbol{Z}$$

Since we can approximate the true conditional distributions, it follows that:

$$X_i \perp_{\theta_x, \theta_y} Y_j|\boldsymbol{Z} \implies X_i \perp Y_j|\boldsymbol{Z} \implies X_i \perp_{\mathcal{G}} Y_j|\boldsymbol{Z}$$

Thus, the graph $\mathcal{G}$ can be identified from the observational data. $\square$

### B.2  PROOF OF LEMMA 2

*Proof.* Let $< \boldsymbol{U}, \mathcal{G}, P >$ be the sequential BN composed of the events from the multi-labeled sequence $S_l = (\{(t_1, x_1, \cdots, (t_L, x_L)\}_{i=1}^{L}, (\boldsymbol{y}_L, t_L))$. Following the temporal precedence assumption A1, the labels $\boldsymbol{y}_L$ can only be caused by past events $(x_1, \cdots, x_L)$, moreover by definition labels does cause any other labels. Thus, $Y_j$ has no descendants, so no children and spouses. Therefore, together with the Markov Assumption we know that $\forall X \in \{\boldsymbol{U} - Pa(Y_j)\} : Y_j \perp X|Pa(Y_j)$. Which is the definition of the MB (Def. 3). Thus, $\textbf{MB}(Y_j) = Pa(Y_j)$.

$\square$

### B.3  PROOF OF THEOREM 1.

*Proof.* By recurrence over the sequence length $L$ of the multi-label sequence $S_l^k$, we want to show that under temporal precedence A1, bounded lagged effects A4, causal sufficiency A2, Oracle Models A3 and using an estimation of the CMI (equation 8) we can identify conditional independence so the Markov Boundary of label $Y_j$ can be identified in the causal graph $\mathbb{G}$.

Let's define $\mathcal{M}_j^L$ as the estimated Markov Boundary of $Y_j$ after observing $L$ events.

**Base Case: $L = 1$:** Consider the BN for step $L = 1$ following the Markov assumption Pearl (1988) with two nodes $X_1, Y_j$. Using $\text{Tf}_x, \text{Tf}_y$ as Oracle Models A3, we can express the conditional probabilities for any node $U$:

$$P(U|\text{Pa}(U)) = \begin{cases} P(X_1) = P_{\theta_x}(X_1|[CLS]) \text{ if } U \in \boldsymbol{X} \\ P(Y_j|X_1) = P_{\theta_y}(Y_j|X_1) \text{ otherwise} \end{cases} \tag{12}$$

Assuming that P is faithful (A2) to $\mathbb{G}$, no hidden confounders bias the estimate (A2) and temporal precedence (A1), using equation 8, we can estimate the CMI such that iif $I(Y_j, X_1|\emptyset) > 0 \Leftrightarrow X_1 \not\perp_{\theta_x,\theta_y} Y_j \implies X_1 \not\perp_{\mathbb{G}} Y_j$ (Lemma 1).

Since we assume temporal precedence A1, we can orient the edge such that $X_1$ must be a parent of $Y_j$ in $\mathbb{G}$. Using Lemma 2, we know that $Par(Y_j) = \textbf{MB}(Y_j) \implies X_1 \in \textbf{MB}(Y_j)$, thus we must include $X_1$ in $M_j^1$, otherwise not.

**Heredity:** For $L = i$, we obtained $M_j^i$ with the sequential BN up to step $L = i$. Now for $L = i + 1$, the sequential BN has $i + 2$ nodes denoted as $\boldsymbol{U'} = (X_1, \cdots, X_i, X_{i+1}, Y_j)$. Using the Oracle Models A3 and following the Markov assumption (Pearl, 1988), we can estimates the following conditional probabilities for any nodes $U \in \boldsymbol{U'}$:

$$P(U|\text{Pa}(U)) = \begin{cases} P(Y_j|\text{Pa}(Y_j)) \approx P_{\theta_y}(Y_j|\text{Pa}(Y_j)), & \text{if } U \in \boldsymbol{Y} \\ P(X|\text{Pa}(X)) \approx P_{\theta_x}(X|\text{Pa}(X)), & \text{otherwise.} \end{cases} \tag{13}$$

By bounded lagged effects (A4) we know that the causal influence of past $X_{\leq i}$ on $Y_j$ has expired. Moreover, no hidden confounders (A2) bias the independence testing. Finally, using equation 8, we can estimate the CMI such that iif $I(Y_j, X_{i+1}|\boldsymbol{Z}) > 0 \Leftrightarrow X_{i+1} \not\perp_{\theta_x,\theta_y} Y_j|\boldsymbol{Z} \implies X_{i+1} \not\perp_{\mathbb{G}} Y_j|\boldsymbol{Z}$ (Lemma 1).

Since we assume temporal precedence A1, we can orient the edge so that $X_{i+1}$ must be a parent of $Y_j$ in $\mathbb{G}$. Using Lemma 2, we know that $Par(Y_j) = \textbf{MB}(Y_j) \implies X_{i+1} \in \textbf{MB}(Y_j)$. Thus $X_{i+1} \in M_j^{i+1}$ which represent the $\textbf{MB}(Y_j)$ for step $i + 1$.

Finally, $\mathcal{M}_j^{i+1}$ still recovers the Markov Boundary of $Y_j$ such that

$$\forall U \in \{\boldsymbol{U'} - \mathcal{M}_j^{i+1}\}, Y_j \perp U|\mathcal{M}_j^{i+1}$$

$\square$

## C  EVALUATION

### C.1  METRICS

The Precision, Recall, and F1-Score for Markov boundary estimation were computed as follows using the True set as the error pattern rule (True Markov Boundary) and the Inferred Markov Boundary set from OSCAR:

- **Precision** ($P$) measures the proportion of correctly identified causal events among all inferred events:

$$P = \frac{|\text{Inferred} \cap \text{True}|}{|\text{Inferred}|}$$

  where $|\text{Inferred} \cap \text{True}|$ is the number of true positive causal events, and $|\text{Inferred}|$ is the total number of inferred causal events.

- **Recall** ($R$) captures the proportion of correctly identified causal events among all true causal events:

$$R = \frac{|\text{Inferred} \cap \text{True}|}{|\text{True}|}$$

  where $|\text{True}|$ is the total number of true causal tokens.

- **F1-Score** ($F_1$) is the harmonic mean of precision and recall, providing a balanced measure:

$$F_1 = \frac{2 \cdot P \cdot R}{P + R}$$

## C.2 PYCAUSALFS

Local structure learning algorithms were all used with $\alpha = 0.1$ in the associated code: `https://github.com/wt-hu/pyCausalFS/tree/master/pyCausalFS/LSL`.

## C.3 MI-MCF

MI-MCF Ma et al. (2025) was used for comparison following the official implementation at `https://github.com/malinjlu/MI-MCF` we used $\alpha = 0.05, L = 268, k_1 = 0.7, k_2 = 0.1$.

# D ABLATIONS

## D.1 NADEs QUALITY.

We did several ablations on the quality of the NADEs and their impact on the one-shot causal discovery phase. In particular, Table 3 presents multiple $\text{Tf}_x, \text{Tf}_y$ with respectively 90 and 15 million parameters or 34 and 4 million parameters. We also varied the context window (conditioning set $Z$), trained on different amounts of data (Tokens) and reported the classification results on the test set of $\text{Tf}_y$ alone. We didn't output the Running time since it was always the same for all NADEs: 1.27 minutes of 50,000 samples and 0.14 for 5000.

Table 3: Ablations of the performance of Phase 1 (One-shot **MB** retrieval) in function of different NADEs with $n = 50,000$ and $n = 500$ samples averaged over 5-folds. Classification metrics use weighted averaging. Metrics are given in %.

| Tokens | Parameters | Context | Precision (↑) | Recall (↑) | F1 Score (↑) | Tfy F1 (↑) |
|--------|-----------|---------|---------------|------------|--------------|------------|
| *For $n = 50,000$ samples* | | | | | | |
| 1.5B | 105m | $c = 4$ | $47.95 \pm 1.05$ | $30.65 \pm 0.51$ | $37.39 \pm 0.67$ | 88.6 |
| 1.5B | 105m | $c = 12$ | $54.62 \pm 1.03$ | $29.88 \pm 0.73$ | $38.63 \pm 0.85$ | 90.43 |
| 1.5B | 105m | $c = 15$ | $\mathbf{55.26 \pm 1.42}$ | $31.37 \pm 0.82$ | $\mathbf{40.02 \pm 1.03}$ | 90.57 |
| 1.5B | 105m | $c = 20$ | $49.52 \pm 1.59$ | $\mathbf{31.76 \pm 0.85}$ | $36.54 \pm 1.10$ | 91.19 |
| 1.5B | 105m | $c = 30$ | $36.65 \pm 1.18$ | $22.75 \pm 0.78$ | $26.57 \pm 0.91$ | $\mathbf{92.64}$ |
| 300m | 47m | $c = 20$ | $39.49 \pm 1.77$ | $26.30 \pm 0.89$ | $29.01 \pm 1.10$ | 83.6 |
| *For $n = 500$ samples* | | | | | | |
| 1.5B | 105m | $c = 12$ | $54.84 \pm 4.55$ | $\mathbf{31.45 \pm 2.23}$ | $\mathbf{39.95 \pm 2.83}$ | 90.43 |
| 1.5B | 105m | $c = 15$ | $55.04 \pm 3.36$ | $29.90 \pm 1.78$ | $38.74 \pm 2.24$ | 90.57 |
| 1.5B | 105m | $c = 20$ | $48.84 \pm 4.01$ | $\mathbf{31.65 \pm 2.37}$ | $36.19 \pm 2.65$ | $\mathbf{91.19}$ |
| 300m | 47m | $c = 20$ | $38.23 \pm 2.91$ | $25.31 \pm 2.39$ | $27.92 \pm 2.25$ | 83.6 |

## D.2 PROPOSAL

We performed an ablation (Tab 4) on the effect of sampling methods to estimate the expected value over all possible context $Z$. We used one A10 GPU on a sample of the test dataset (4000 random samples) composed of 205 labels with a batch size of 4 during inference. We tested top-k sampling with $k = \{20, 35\}$ Fan et al. (2018) with and w/o a temperature scaler of $T$ to log-probabilities $\hat{x}$ such that

$$\hat{x}' = \text{softmax}(\log \hat{x}/T)$$

And a combination of top-k and a top-nucleus sampling Holtzman et al. (2020) with different probability mass $p = \{0.8, 1.2\}$ and finally a permutation of token position within the context c. We

fixed a dynamic threshold with z score $k = 3$ and performed 10 runs. Then, we reported the average and standard deviation of each classification metric and elapsed time (sec).

Without a surprise, sampling increases the predictive performance of OSCAR by a large margin. More interestingly, different sampling types have different effects on specific averaging. This has a 'smoothing' effect on the CMI curve when multiple labels are present in the sequence. When having no upsampling, the sensitivity of the CMI of different labels is increased, which makes it more difficult to capture a threshold and a potential cause. We can notice that globally, top-k sampling provides better results, especially with a combination of top-p=0.8 afterwards. Sampling with the same tokens (*Permutation*) is not a good choice, giving more diversity by sampling from the next-event prediction $Tf_x$ yielded better results. We will choose **Top-k+p=0.8** for the increased F1 Micro and high F1 Macro, and Weighted.

Table 4: One-shot Classification performance and Elapsed Time (sec) across different sampling methods. Best results are shown in **Bold** and Best ex aequo in underline.

| Proposal | F1 Micro (%) | F1 Macro (%) | F1 Weighted (%) | Time (sec) |
|---|---|---|---|---|
| w/o Sampling | 14.07 | 12.29 | 16.67 | **49.30 ± 0.30** |
| Permutation | 18.22 ± 0.36 | 13.75 ± 0.09 | 19.21 ± 0.03 | 557.82 ± 0.13 |
| Top-k=20 | 26.77 ± 0.71 | 23.83 ± 0.19 | 29.25 ± 0.07 | 557.4 ± 0.13 |
| Top-k=35 | 26.57 ± 0.96 | 24.08 ± 0.23 | 29.30 ± 0.07 | 557.35 ± 0.10 |
| Top-k=35+T=0.8 | 27.36 ± 0.65 | 23.77 ± 0.21 | 28.98 ± 0.07 | 557.45 ± 0.11 |
| Top-k=35+T=1.2 | 26.59 ± 1.49 | **24.62 ± 0.29** | **29.52 ± 0.06** | 557.45 ± 0.12 |
| Top-k=25+p=0.8 | 27.98 ± 0.67 | 23.82 ± 0.28 | 29.18 ± 0.07 | 558.07 ± 0.07 |
| **Top-k=35+p=0.8** | **28.82 ± 0.75** | 24.06 ± 0.25 | 29.17 ± 0.07 | 558.16 ± 0.14 |
| Top-k=35+p=0.9 | 26.39 ± 0.99 | 24.12 ± 0.31 | 29.26 ± 0.11 | 558.11 ± 0.12 |
| Top-k=35+p=0.9+T=0.9 | 27.63 ± 0.75 | 23.90 ± 0.24 | 29.04 ± 0.09 | 558.07 ± 0.12 |
| Top-k=35+p=0.9+T=1.1 | 26.75 ± 1.30 | **24.47 ± 0.24** | 29.45 ± 0.09 | 558.06 ± 0.11 |

## D.3 SAMPLING NUMBER

We experimented with different numbers of $N$ for the sampling method across different averaging (micro, macro, weighted), Fig .4. We performed 8 different runs and reported the average, standard deviation and elapsed time. We can say that generally, sampling with a bigger $N$ tends to decrease the standard deviation and give more reliable Markov Boundary estimation. Moreover, as we process more samples, the model is gradually improving at a logarithmic growth until it converges to a final score. We also verify that our time complexity is linear with the number of samples $N$. Based on these results, we choose generally $N = 68$ as the number of samples.

## D.4 DYNAMIC THRESHOLDING

We performed ablations on the effect of $k$ during the dynamic thresholding of the CMI (equation 10) to access conditional independence in Fig .5. To balance the classification metrics across the different averaging, we set $k = 2.75$.

Figure 4: Evolution of several classification metrics (one-shot) and elapsed time per sample in function of the number of samples $N$ chosen. Results are reported using 1-sigma error bar.

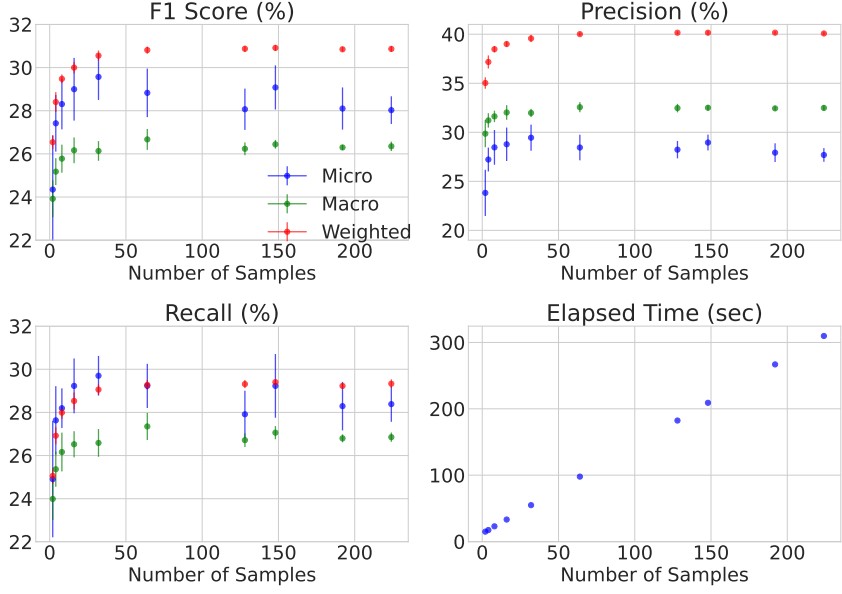

Figure 5: Evolution of one-shot F1 Score, Precision and Recall in function of coefficient $k$. Results are reported using 1-sigma error bar.

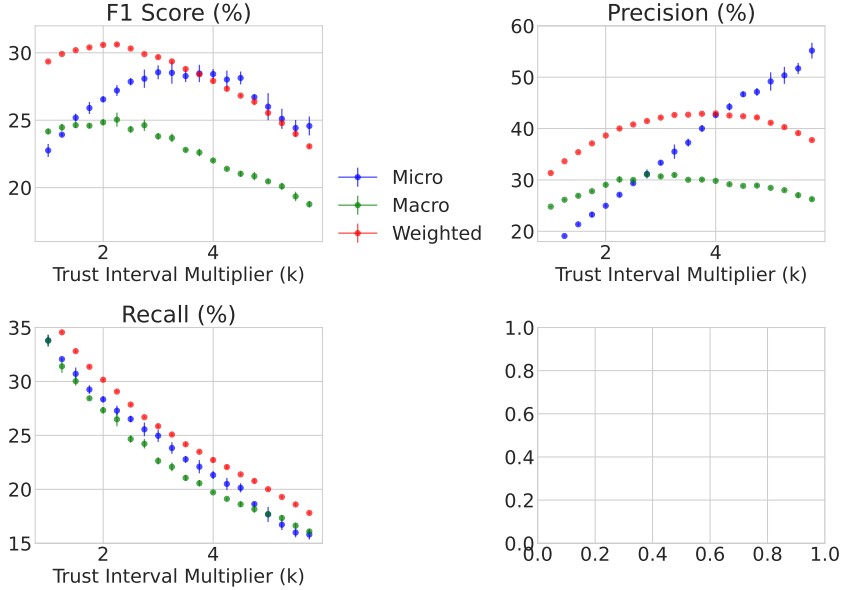

# E    EXTENDED DISCUSSION ON ASSUMPTIONS

Our approach relies on several assumptions that enable one-shot causal discovery under practical and computational constraints.

**Temporal Precedence** Temporal precedence (A1) simplifies directionality and faithfulness to $\mathbb{G}$. It allows for instantaneous influence, which aligns better with log-based data in cybersecurity or vehicle diagnostics, where events can co-occur at the same timestamp. However, this places strong reliance on precise event time-stamping. Even though we only test $X_i \rightarrow Y_j$, this could falsified the conditioning test $\mathbf{Z}$.

**Bounded Lagged Effects.** The bounded lagged effects (A4) assumption enables us to restrict causal influence and recover the **MB** of each label using Theorem 1. It also makes the computation faster. In most real-world sequences where relevant history is limited, this holds empirically. Nonetheless, in highly delayed causal chains, like financial transactions, some influence may be missed.

**Causal Sufficiency.** As with many causal discovery approaches, we assume all relevant variables are observed (A2). Although it sounds like a strong assumption, interestingly, in high-cardinality domains such as vehicle diagnostics, the volume of recorded events may reduce but not eliminate the risk of hidden confounding.

**Inter-label Effects.** By definition, the labels are explained solely by events. While simplifying multi-label causal discovery, this intrinsic assumption could be relaxed in future work by using the *do* operator Pearl (2009) to perform interventions on common causal variables of multiple labels. For exemple, our current framework estimates the Markov Boundaries for each label independently. However, inter-label dependencies can exist, particularly when labels share overlapping Markov Boundaries (e.g $MB_1 = [X_1, X_3], MB_2 = [X1, X_2]$. We propose to investigate a 'Phase 2' for OSCAR, focusing on inter-label dependencies through simulated interventions. For instance, if we consider a sequence $S_1$ of two labels $Y_1, Y_2$ with the MB above, we could perform counterfactual interventions by applying $do(X_1 = 0), do(X_3 = 0)$to $S_1$. Then we would observe the average change in the likelihood of $Y_1$ which if it is non-zero, would indicate a dependence between $Y_1$ and $Y_2$. Wu et al. (2020) points out that the assumptions of these inter-label dependencies are already anchored in the Markov Boundaries, we do the same here.

**NADEs.** Due to the usage of flexible NADEs, we can relax common assumptions regarding data generation processes such as Poisson Processes or SCMs. Finally, as seen in the Ablations D.1, the effectiveness of OSCAR hinges on the capacity of $Tf_x$ and $Tf_y$ to approximate true conditional probabilities (A3) and provide Oracle CI-test. While assuming Oracle tests is common in the literature Xie et al. (2024); Li et al. (2016) and necessary to recover correct causal structures, this remains a strong assumption. And it is only valid to the extent that the models are perfectly trained. Especially for multi-label classification, performance may degrade in underrepresented regions of the data distribution.

For example, we analyze on a reduced dataset, the performance of OSCAR in function of the **MB** length:

Figure 6: Evolution of the One-Shot Recall, Precision and F1-Score in function of the Markov Boundary length $|\mathbf{MB}(Y_j)|$ using $n = 45969$ samples.

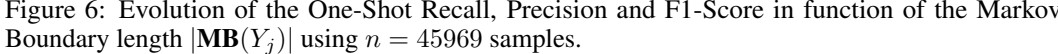
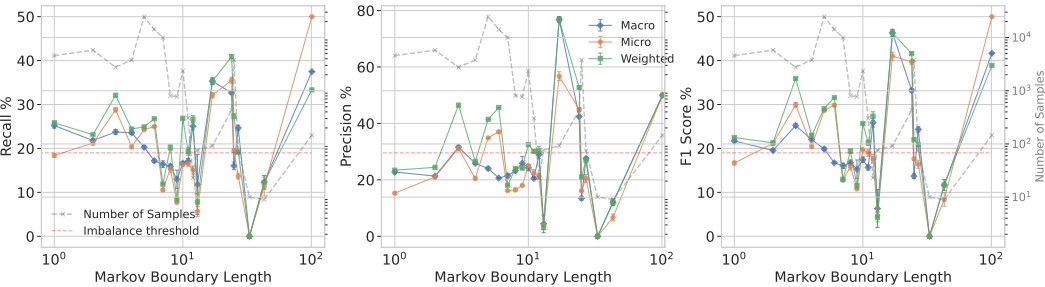

Figure 6 reveals the classification performance depending on the number of nodes in the ground truth **MB**. On the same plot is drawn in grey the number of samples that each **MB** length contains (to account for imbalance). We observe that generally, a bigger $|\mathbf{MB}(Y_j)|$ does not imply a reduction in performance, highlighting the capability of OSCAR to retrieve complex Markov Boundaries in high-dimensional data. However, we observe that past a certain number of samples (imbalance threshold in red $\approx 7 \times 10^2$ samples), the classification metrics are directly correlated with the number of samples per $|\mathbf{MB}(Y_j)|$. This indicates that $Tf_x, Tf_y$ struggle to output proper conditional probabilities, which deteriorates the CI-test when having rare classes. Therefore, when using OSCAR and more generally assumption A3, one should carefully assess class imbalance in the pretraining phase.

# F  FIGURES

## F.1  EXPLAINATION EXAMPLE

To enhance interpretability and illustrate the learned relationships, we present graphical explanations of error pattern occurrences based on sequences of Diagnostic Trouble Codes (DTCs). For each case, we selected representative samples that reflect diverse yet intuitive failure scenarios.

Fig. 8 depicts a clear-cut example involving a single failure label related to the emergency antenna system. In contrast, Fig. 9 captures a more intricate case where airbag and tire pressure (RDC) malfunctions co-occur. These graphs highlight the influence of preceding events, with causal contributions shown in orange and red, and inhibitory effects illustrated in pink. Such visualisations serve to provide both human-understandable insights and support for the model's reasoning process.

Figure 7: Example of a sequence of events (DTCs) that lead to a steering wheel degradation and a power limitation as outcome labels. The inhibitory strengths are shown in violet and causal strengths in orange and red depending on the magnitude.

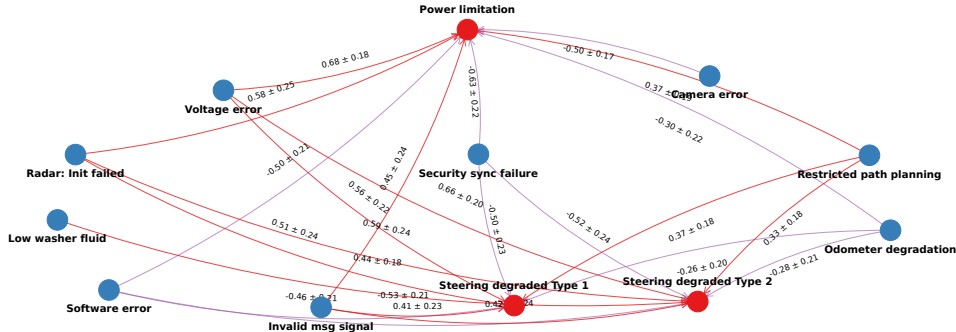

Figure 8: Example of a sequence of events (DTCs) that lead to an emergency antenna dysfunction as outcome labels. The inhibitory strengths are shown in pink and causal strengths in orange and red

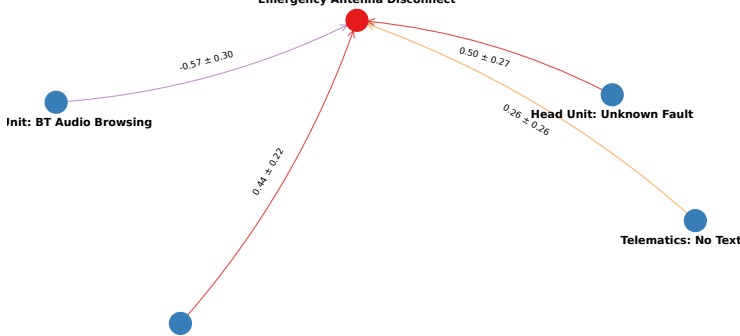

Figure 9: Example of a sequence of events (DTCs) that lead to an airbag and tire pressure malfunctions as outcome labels. The inhibitory strengths are shown in pink and causal strengths in orange and red

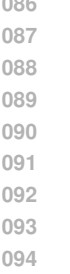

Figure 10: Example of a sequence of events (DTCs) that lead to a steering wheel degradation and a power limitation as outcome labels. The inhibitory strengths are shown in violet and causal strengths in orange and red depending on the magnitude.



## G  IMPLEMENTATION

The following is the full implementation of OSCAR in PyTorch Paszke et al. (2019).

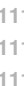

```python
def topk_p_sampling(z, prob_x, c: int, n: int = 64, p: float = 0.8, k:
    int = 35,
                    cls_token_id: int = 1, temp: float = None):
    # Sample just the context
    input_ = prob_x[:, :c]

    # Top-k first
    topk_values, topk_indices = torch.topk(input_, k=k, dim=-1)

    # Top-p over top-k values
    sorted_probs, sorted_idx = torch.sort(topk_values, descending=True,
    dim=-1)
    cum_probs = torch.cumsum(sorted_probs, dim=-1)
    mask = cum_probs > p

    # Ensure at least one token is kept
    mask[..., 0] = 0

    # Mask and normalize
```

```
18     filtered_probs = sorted_probs.masked_fill(mask, 0.0)
19     filtered_probs += 1e-8  # for numerical stability
20     filtered_probs /= filtered_probs.sum(dim=-1, keepdim=True)
21
22     # Unscramble to match the original top-k indices
23     # Need to reorder the sorted indices back to the original top-k
24     reorder_idx = torch.argsort(sorted_idx, dim=-1)
25     filtered_probs = torch.gather(filtered_probs, -1, reorder_idx)
26
27     batched_probs = filtered_probs.unsqueeze(1).repeat(1, n, 1, 1)
        # (bs, n, seq_len, k)
28     batched_indices = topk_indices.unsqueeze(1).repeat(1, n, 1, 1)
        # (bs, n, seq_len, k)
29
30     sampled_idx = torch.multinomial(batched_probs.view(-1, k), 1)
        # (bs*n*seq_len, 1)
31     sampled_idx = sampled_idx.view(-1, n, c).unsqueeze(-1)
32
33     sampled_tokens = torch.gather(batched_indices, -1, sampled_idx).
       squeeze(-1)
34     sampled_tokens[..., 0] = cls_token_id
35
36     # Reconstruct full sequence
37     z_expanded = z.unsqueeze(1).repeat(1, n, 1)[..., c:]
38     return torch.cat((sampled_tokens, z_expanded), dim=-1)
39
40 from torch import nn
41 def OSCAR(tfe: nn.Module, tfy: nn.Module, batch: dict[str, torch.Tensor],
        c: int, n: int, eps: float=1e-6, topk: int=20, k: int=2.75, p=0.8)
       -> torch.Tensor:
42     """ tfe, tfy: are the two autoregressive transformers (event type and
        label)
43         batch: dictionary containing a batch of input_ids and
       attention_mask of shape (bs, L) to explain.
44         c: scalar number defining the minimum context to start inferring,
        also the sampling interval.
45         n: scalar number representing the number of samples for the
       sampling method.
46         eps: float for numerical stability
47         topk: The number of top-k most probable tokens to keep for
       sampling
48         k: Number of standard deviations to add to the mean for dynamic
       threshold calculation
49         p: Probability mass for top-p nucleus
50     """
51     o = tfe(attention_mask=batch['attention_mask'], input_ids=batch['
       input_ids'])['prediction_logits'] # Infer the next event type
52     x_hat = torch.nn.functional.softmax(o, dim=-1)
53
54     b_sampled = topk_p_sampling(batch['input_ids'], x_hat, c, k=topk, n=n
       , p=p) # Sampling up to (bs, n, L)
55     n_att_mask = batch['attention_mask'].unsqueeze(1).repeat(1, n, 1)
56
57     with torch.inference_mode():
58         o = tfy(attention_mask=n_att_mask.reshape(-1, b_sampled.size(-1))
       , input_ids=b_sampled.reshape(-1, b_sampled.size(-1))) # flatten and
       infer
59         prob_y_sampled = o['ep_prediction'].reshape(b_sampled.size(0), n,
        batch['input_ids'].size(-1)-c, -1) # reshape to (bs, n, L-c)
60
61         # Ensure probs are within (eps, 1-eps)
62         prob_y_sampled = torch.clamp(prob_y_sampled, eps, 1 - eps)
63
64         y_hat_i = prob_y_sampled[..., :-1, :] # P(Yj|z)
65         y_hat_iplus1 = prob_y_sampled[..., 1:, :] # P(Yj|z, x_i)
```

```
66
67          # Compute the CMI & CS and average across sampling dim
68          cmi = torch.mean(y_hat_iplus1*torch.log(y_hat_iplus1/y_hat_i)+
        (1-y_hat_iplus1)*torch.log((1-y_hat_iplus1)/(1-y_hat_i)), dim=1)
69          # (BS, L, Y)
70          cs = y_hat_iplus1 - y_hat_i
71          cs_mean = torch.mean(cs, dim=1)
72          cs_std = torch.std(cs, dim=1)
73
74          # Confidence interval for threshold
75          mu = cmi.mean(dim=1)
76          std = cmi.std(dim=1)
77          dynamic_thresholds = mu + std * k
78
79          # Broadcast to select an individual dynamic threshold
80          cmi_mask = cmi >= dynamic_thresholds.unsqueeze(1)
81
82          cause_token_indices = cmi_mask.nonzero(as_tuple=False)
83          # (num_causes, 3) --> each row is [batch_idx, position_idx,
        label_idx]
84          return cause_token_indices, cs_mean, cs_std, cmi_mask
```

**Remark.** *Since tfy contains tfe as backbone, in practice we need only one forward pass from tfy and extract also $\hat{x}$, so tfe is not needed. We let it to improve understanding and clarity.*

