# OpenReview forum: "One-Shot Multi-Label Causal Discovery in High-Dimensional Event Sequences"
_ICLR.cc/2026/Conference — ICLR 2026 Conference Withdrawn Submission_

### Official Review · Reviewer_tfnt · 2025-10-30

**Soundness:** 2
**Presentation:** 3
**Contribution:** 2
**Rating:** 2
**Confidence:** 4

**Summary:**

This paper introduces OSCAR, a method for one-shot, multi-label causal discovery in high-dimensional event sequences. The core idea is to use two Transformer models as density estimators to compute Conditional Mutual Information (CMI) and identify the Markov Boundary for each label from a single sequence. The authors position their work as a solution to the scalability issues of traditional constraint-based and Granger-style methods. They validate OSCAR on a large-scale vehicular dataset, demonstrating significant improvements in speed and the ability to handle thousands of event types and hundreds of labels.

**Strengths:**

Strengths

- The paper addresses an important practical problem for learning causal structure from a high-dimensional event sequence.

- The use of a large-scale, real-world vehicular dataset for evaluation shows the promising performance of the proposed method.

**Weaknesses:**

Weaknesses

- The goal of this paper is to discover the learning structure from event sequences, which is a central task in many point process-based methods. However, this paper positions itself against the traditional constraint-based methods, which overlook a branch of literature in the temporal point process.

- The paper emphasizes its "one-shot" capability as a key contribution. However, most causal discovery methods are inherently "one-shot" in the sense that they infer structure from a given dataset (which can be one long sequence). This claim feels overstated without a clear distinction of what "one-shot" means in this context compared to the standard setup of other methods.

- It also lacks of discussion of the motivation for using a Transformer-based architecture. It would be better to discuss what is beneficial and provide a clear analysis of why the Transformer is optimal for this specific causal task.

- The experiment comparison is limited to the constraint-based methods. A more rigorous and convincing evaluation would include comparisons with scalable point process-based causal discovery methods and other recent neural-based approaches.

- As noted, the citation style is inconsistent. In-text citations should be enclosed in parentheses (e.g., (Author et al., Year)) for better readability.

**Questions:**

See the weakness above.

---

### Official Review · Reviewer_PL68 · 2025-10-30

**Soundness:** 3
**Presentation:** 3
**Contribution:** 3
**Rating:** 6
**Confidence:** 3

**Summary:**

This paper presents OSCAR, a novel one-shot multi-label causal discovery framework designed for high-dimensional event sequences. The method leverages two pretrained Transformer-based autoregressive density estimators to compute conditional mutual information (CMI), allowing direct identification of Markov Boundaries for each label from a single observed sequence. The authors provide theoretical identifiability guarantees and validate the approach on a large-scale real-world vehicular dataset involving over 29,000 event types and 474 labels. OSCAR demonstrates significant scalability and interpretability advantages over traditional constraint-based methods.

**Strengths:**

1.	The paper tackles an important and underexplored problem from a one-shot perspective, i.e., causal discovery in high-dimensional, multi-label event sequences.
2.	The authors clearly articulate key assumptions, including temporal precedence, causal sufficiency, bounded lagged effects, and derive identifiability results that support the validity of their approach.
3.	Experiments on a large real-world vehicle diagnostic dataset in this paper provide convincing evidence of the method’s effectiveness and real-world applicability.
4.	The comparisons to existing causal discovery baselines, despite their scalability limitations, highlight OSCAR’s efficiency advantage.

**Weaknesses:**

1.	Some baseline methods in the experiments fail to execute properly on high-dimensional data, which weakens the strength of the comparative analysis. Conducting additional experiments on smaller or synthetic datasets where all methods can be executed would make the empirical evaluation more comprehensive and convincing.
2.	While the appendix mentions ablations, the main text lacks detailed discussion on the effect of hyperparameters such as context length, sample count, and threshold. A concise sensitivity analysis would help establish robustness.
3.	The assumptions of causal sufficiency and bounded lagged effects may not hold in many real-world applications, e.g., healthcare or finance, where hidden confounders and delayed effects are common. The paper could discuss potential violations of these assumptions and how they might affect inference quality.
4.	In the context of this paper, qualitative validation refers to evaluating the interpretability and real-world relevance of the causal graphs discovered by the method. While the paper focuses on quantitative performance metrics, e.g., precision, recall, F1-score, there is limited discussion on how well the discovered causal relationships align with domain-specific knowledge or expert understanding.

**Questions:**

1.	Some baseline methods in the experiments fail to execute properly on high-dimensional data, which weakens the strength of the comparative analysis. Conducting additional experiments on smaller or synthetic datasets where all methods can be executed would make the empirical evaluation more comprehensive and convincing.
2.	While the appendix mentions ablations, the main text lacks detailed discussion on the effect of hyperparameters such as context length, sample count, and threshold. A concise sensitivity analysis would help establish robustness.
3.	The assumptions of causal sufficiency and bounded lagged effects may not hold in many real-world applications, e.g., healthcare or finance, where hidden confounders and delayed effects are common. The paper could discuss potential violations of these assumptions and how they might affect inference quality.
4.	In the context of this paper, qualitative validation refers to evaluating the interpretability and real-world relevance of the causal graphs discovered by the method. While the paper focuses on quantitative performance metrics, e.g., precision, recall, F1-score, there is limited discussion on how well the discovered causal relationships align with domain-specific knowledge or expert understanding.

---

### Official Review · Reviewer_WwYD · 2025-10-31

**Soundness:** 3
**Presentation:** 2
**Contribution:** 2
**Rating:** 4
**Confidence:** 3

**Summary:**

This paper studies causal discovery on high-dimensional event sequences. The setting of one-shot multi-label causal discovery is considered, and a transformer-based model is designed to estimate the conditional mutual information. The causal discovery performance and the running time efficiency are evaluated on a real-world dataset.

**Strengths:**

1. It is a good contribution to release the real-world dataset used in the experiment.

2. The proposed method is computationally efficient.

**Weaknesses:**

1. How to extend the proposed method to the situation with hidden confounders?

2. Is it possible to speed up the baselines using the similar tricks of the proposed method?

3. To evaluate the performance and the robustness of the proposed method, it would be better to conduct experiments on more datasets.

4. It is surprising that all the baselines perform terribly on the dataset. What is the reason behind this observation?

5. Due to the heavy computational cost, the baselines cannot be completely conducted on the entire dataset. It would be better to compare them on a smaller dataset. Otherwise, the performance advantage of the proposed method is not convincing.

**Questions:**

Please refer to the weaknesses.

---

### Official Review · Reviewer_StPU · 2025-11-07

**Soundness:** 3
**Presentation:** 2
**Contribution:** 2
**Rating:** 4
**Confidence:** 3

**Summary:**

This paper proposes OSCAR, a framework for one-shot causal discovery in high-dimensional event data. The methodology uses pre-trained transformer-based architectures, and identifies computes markov boundaries for labels via conditional mutual information between past and current events. The authors present experiments on high-dimensional real-world data, and compare their method with other constraint-based methodologies.

**Strengths:**

- One-shot causal discovery is a very interesting and challenging problem.
- The experiments demonstrate that the method is efficient in comparison to other constraint-based methodologies.
- Using Markov Boundaries rather than the full causal graph is a much more efficient way to approach causal discovery for such high-dimensional settings.
- Evaluation on real-world settings is far more convincing than synthetic data in terms of the applicability of the presented methodology.
- Experiments are detailed, with extensive details for reproduction.

**Weaknesses:**

**Assumptions**

The assumption on oracle models seems quite strong. It might be a good idea to provide some intuition or references on why this is reasonable.

**Presentation and clarity**

The methodology presented is very promising. However, I consider the paper could benefit with some revisions in terms of clarity and presentation, especially Section 4.1. Please find below some comments:
- All the citations use \citet. Please use \citep accordingly as in most of the cases you require the citations in brackets.
- For the causal discovery crowd, MBs are not usual, and I believe it would be important to stress the justification of targeting MBs rather than the full causal graph.
  -  Line 53: "Specifically, we seek to extract, for each label, the minimal set of causal events—its Markov Boundary."
    - Here you present MBs, please provide motivation for pursuing MBs here.
- Line 163: Why $X_i^{t_i}$, but $Y_{i+1}^{t_{i+1}}$. Considering this, why Figure 1 only shows Y_L?
- Figure 2 is not referenced in the main text. Given it provides very important details about the core elements in the methodology, I suggest the procedure detailed in the figure should be explained when introducing equations. For example, it would help understand how sampling from Tf_x works.
- Eq. (5), second Z should be bold right?
- Eq. (7) uses Y_j, X_i, and Z. However, in Equations (8), (9), and the one in line 304 use X_{i+1}. Is this a typo? Where does X_{i+1} come from?
- Section 4.1.1 reads confusing as the text is mixed with pseudocode. Please, consider wrapping the section together with one "Algorithm" block instead. Furthermore, it seams to be repeating indications on section 4.1.1.
- Figure 3 only shows its caption. Is this expected?

**Methodology lacks details on sampling**
- In Line 294: Samples $z^{(l)}$ are drawn from Tf_x. However, Tf_x computes p(x_i | z). I am assuming that for this you need to do ancestral sampling or similar (sequentially sampling z^{(l)}), and repeat N times. Could you provide more details of how these samples are computed?
- If In Eq. (7) the expectation considers x_i, why don't we take samples x_i's as well? (only z's are sampled according to the text)

**Comments on experiments**
- The ground truth MB is not obtained from a direct causal graph, but is derived from boolean rules. Furthermore, the rules vary in complexity, with some missing on the dataset. It is not clear to me that rules and causation are the same in this context. Considering this, it would be better to showcase at least one small synthetic experiment, where grountruth is fully controllable.
- Tables 1 and 2 are not very informative on other baselines. A single line explaining long runtimes suffices.
- The data for baselines is converted into a multi-one-hot data-frame. This causes that for even shorter sequences, the CI test fails. This seems like the baselines are at a disadvantage due to the treatment of the data. I would recommend exploring the following.
  - Reducing samples will not do the trick for the baselines. Would it be possible to find a reduced set of labels and samples where at least the baselines can produce some MBs?
  - Would it be possible to run time series causal discovery baselines? PCMCI, Granger-Lasso, DYNOTEARS, Rhino, etc. It feels like the current baselines are at a massive disadvantage and it would improve the paper to provide experiments with other flexible baselines as above.
- The model with n=500 seems to perform similarly with n=50000, although it takes x100 more to run. Would it be possible to have a plot to observe performance in terms of sequences n?

I am very happy to raise my score if my comments on the clarity and methodology are considered in rebuttal.

**Questions:**

See Above.

---

### Note · Authors · 2025-11-18

I have read and agree with the venue's withdrawal policy on behalf of myself and my co-authors.